# Rapid assessment of noma: Reporting on forgotten and neglected disease in Ethiopia

Wendemagegn Enbiale[1,2]*

1 Bahir Dar University, College of Medicine and Health Sciences, Bahir Dar, Ethiopia, 2 Collaborative Research and Training Center for Neglected Tropical Diseases, Arba Minch University, Arba Minch, Ethiopia

* wendemagegnenbiale@gmail.com

## Abstract

### Background

Noma is a rapidly progressing, invasive, and debilitating orofacial disease that primarily affects the most vulnerable and marginalised populations worldwide. The highest- risk group includes pre-school children, exposed to other risk factors, such as malnutrition and poverty-related diseases. Since 2010, Ethiopia has reported an increasing number of noma cases, primarily identified through medical missions. Data on the disease burden and epidemiology are essential for planning service delivery and developing effective disease prevention strategies. In this endeavour to document noma.s presence in Ethiopia, and assess the health system capacity for noma care, we have performed a rapid assessment.

### Methodology

We performed a rapid assessment including a desk and literature review, health sector capacity assessment and a retrospective analysis of hospital records to identify all confirmed cases of noma cases from 2015–2022, based on data from NGOs and health facility records.

### Result

The desk review revealed that Ethiopia lacks a national policy on noma. However, the national health policy emphasizes the prevention and control of poverty-related diseases. There is no formal oral health program within the primary healthcare, aside from the limited dental care availability in regional/referral hospitals and private sector. The retrospective assessment has extracted 69 noma cases record reported from January 2015 to December 2020, with 97% of case record came from two NGO's supporting surgical mission. Cases were reported from nearly every region of the country with a notable concentration in Amhara region. The trend of cases being cared has decreased from 2015 to 2020 and no record is found for acute cases of noma.

### Conclusion

The rapid assessment highlights a critical lack of research and surveillance programmes for noma. Efforts to increase public awareness and educate community workers and primary health care professionals on identification of noma and referring patients for care are

**Data Availability Statement:** Data base is attached as supporting document.

**Funding:** The author(s) received no specific funding for this work.

**Competing interests:** The authors have declared that no competing interests exist.

essential. As a first step toward eliminating noma, the disease should be added to the national list of neglected tropical diseases, followed by integrated control programmes through the existing health extension system to expand oral health service.

## Author summary

Noma is a rapidly progressing and debilitating orofacial disease, primarily affecting vulnerable, malnourished children in impoverished regions. Ethiopia has seen increased cases of Noma since 2010, mainly through medical missions. Understanding the disease's burden is essential for planning and prevention.

To assess Noma in Ethiopia, we conducted a rapid evaluation, including literature reviews, health sector assessments, and a retrospective analysis of confirmed cases from 2015–2022. Our findings show that Ethiopia lacks a standalone national policy on Noma, though the national health policy addresses poverty-related diseases. There is also no formal oral health program within primary care; dental services are limited to regional hospitals and the private sector.

The review identified 69 Noma cases reported between 2015 and 2020, mostly from two hospitals in the capital supported by NGOs. Cases were found across nearly all regions, with a concentration in Amhara. No acute Noma cases were recorded, and overall cases declined over the years.

This assessment underscores the need for enhanced public awareness, better training for health workers, and integrated control programs to address Noma through existing health systems.

## Introduction

Noma, also known as cancrum oris or gangrenous stomatitis, is a progressive and debilitating orofacial disease primarily affecting marginalized populations worldwide. Often referred to as the "face of poverty," this non- communicable and invasive condition typically targets children between the ages of 2 and 6, though cases in immunocompromised adults have also been reported [1,2].

While the exact cause of noma remains unclear, it is believed to result from a combination of factors, including malnutrition, immunocompromis (e.g., bacterial or viral infections, immunosuppression from HIV, tuberculosis, leukemia, or non-Hodgkin lymphoma), concomitant diseases like measles or malaria, poor oral hygiene, certain social and environmental factors, damage to the gingival mucosa barrier, and bacterial infections [2–5].

Noma progresses through five stages, with the acute stage encompassing the first three stages, which typically developing over two weeks. If untreated, noma has an estimated mortality rate of 80–90% [1,6,7]. Without timely intervention, the disease advances to necrotising gangrene in the orofacial area (third stage). Survivors of this stage endure wound healing and scarring over several months (fourth stage), which can result in facial deformities, trismus, and ankyloses of bones and joints. The final stage, known as noma sequelae, occurs approximately a year later and is marked by significant functional, visual, and physical disabilities that hinder food intake and carry a heavy social stigma [7].

Historically, noma was reported widely in Europe and India during the 1800s, as well as in concentration camps during World War II and the war-time population of the Netherlands following the winter famine of 1944/45 [8,9]. However, since the latter half of the 20th century, noma cases have predominantly been documented in low and middle-income countries in

Africa, South America, and Asia. Burkina Faso, Ethiopia, Mali, Niger, Nigeria, and Senegal have reported the highest number of cases [9]. Many cases remain undetected and unreported due to various factors, including limited diagnostic skills among health care providers, the rapid disease progression and high acute-phase mortality, a lack of routine surveillance and inclusion health information systems, and family stigma that leads to hiding affected children [10].

With limited access to treatment during the disease active phase, understanding of tnoma remains scarce. The World Health Organization's most recent estimate from 1999, projected approximately 140,000 new noma cases annually worldwide, with 770,000 individualss living with Noma sequelae at that time. However, the origin and reliability of this estimate are unclear [11]. In Senegal, a country-specific estimates have shown an annual incidence of 4.2 acute noma cases per million children, while a recent retrospective chart review in northwest Nigeria estimated a community-based point prevalence of 33 out of every 1000 children aged 0–15 years [12,13].

Ethiopia, despite a high incidence of noma, case documentation has largely been limited to reports from medical missions and non-governmental organizations since 2010 [14,15]. In June 2022, the Ethiopian Ministry of Health has expressed interest in joining the WHO Regional noma Control Program (RNCP) for the "sustainable elimination of noma as a public health problem in the African Region". Comprehensive data on disease burden and epidemiology are important for planning and prioritizing service delivery, as well as for formulating disease prevention strategies.

In this context we conducted a rapid assessment with the following objectives: 1) to understand the burden and geographic distribution of noma 2) to explore stakeholder engagement in noma prevention and control; 3) to assess health professionals' knowledge of noma and 4) to evaluate the health system capacity to respond to noma prevention and control.

## Methodology

### Ethics statement

The assessment was commissioned by the WHO regional office in collaboration with the Ethiopian Federal Ministry of Health (MOH). We secured a support letter from the MOH for the respective hospitals, granting access to the Health Information Management System and outpatient clinical records. Individual consent was not possible to collect since the study is from retrospective data but to ensure privacy and confidentiality, the data was anonymized for any information that could identify participants.

Given that the project was an integral part of the MOH for health system planning, a formal ethical clearance was deemed unnecessary. Nevertheless, Bahir Dar University's Institutional Review Board granted a waiver for the publication of the data, with Protocol number 785/2023, on September 11, 2023.

### Study design

We conducted a concurrent triangulation mixed methods (qualitative and quantitative) a retrospective cohort study, including literature review and health sector capacity assessment to identify confirmed cases of Noma cases based on 2015–2022 data from NGOs and hospital record books.

### General setting

Ethiopia, the second-most populous country in Africa, has a population of approximately 117,000,000, with 20.9% residing in urban areas. Foreign aid plays a significant role in

Ethiopia's GDP, as it is among the top 10 aid recipients globally. Ethiopia is one of the poorest countries, with a per capita income of $960, and the country ranks low in Africa for total health expenditure [16]. According to the National Health Accounts (NHA-7) report, Ethiopia's per capita health expenditure in 2019/20 was $36.3 USD, significantly below the recommended $86 USD required to provide basic services in low-income countries [17].

Currently, the country's potential public health coverage is 100%, with a life expectancy of 64 years. The maternal mortality rate (MMR) stands at 412/100,000, the under-5 mortality rate (U5MR) is 55/1,000, and HIV prevalence is 0.9%. The childhood coverage of all basic vaccines is 43.1%, and 37% of children under 5 are stunted, with regional variations ranging from 12% in Addis Ababa to 49% in Tigray. Ethiopia has the lowest health workforce density in sub-Saharan Africa, with only 1.4 health workers per 1,000 population [16,17]. The top five risks leading to deaths and disabilities in the country include malnutrition, WaSH (Water, Sanitation, and Hygiene), air pollution, dietary risks, and high fasting blood glucose levels [18].

The Health care system in Ethiopia is decentralized, with three tiers. These include, at the primary care level, is a District-based system that includes a Health post, Health centeres and Primary Hospital. The secondary care level includes General Hospitals to referral hospitals, while the tertiary level provides care at teaching and Specialised Hospitals.

## Data collection and analysis

*Quantitative data collection;* For the retrospective data collection, 13 referral hospitals in six regions and two administrative towns were chosen by purposive sampling. From the selected Hospital one is a private hospital and the other 12 are public hospitals (Fig 1 and Table 1).

The selection of hospitals was done based on convenience prioritizing institutions that provide plastic surgery, oral care, and dermatology services in each major regions in the country, except Tigray which is not represented in the study since it was not accessible in the data collection period because of the war. Two of the only NGOs (Project Harar and Facing Africa) involved in Noma management and care in the country have been included in the study.

The selection of hospitals and NGOs was based on convenience, with focus on institutions that provide plastic surgery, oral care, and dermatology services in each region. Potential entry points for Noma cases were identified by consulting healthcare workers, including physicians and dental surgeons, and reviewing health facility records.

Comprehensive data on noma surgical management was gathered from the service record books of the two NGOs. The record ware validated by tracing back the patient files at each hospital where procedure had been performed. All confirmed noma cases, were included in the study. Information collected from the records covered participants' sociodemographic data (name, age, sex, nationality, residence), year of encounter, and any other associated pathologies present at the time of presentation.

A database was created, considering all noma cases documented in hospital registers from 2015 to 2022. Data collection was conducted I person between September 20 to November 2, 2022. Data were initially collected on paper, and then entered into an Excel database for analysis.

The national annual Neglected Tropical disease meeting organized in Bahir Dar (September 19 to 21, 2022) provided an opportunity to recruit participants for assessing health workers knowledge and health facility capacity. Out of approximately 200 attendees, 40 participants were randomly selected for this assessment.

Qualitative data were analysed using descriptive statistics.

*Qualitative data collection;* Qualitative data ware collected through semi-structured interviews with three plastic surgeons involved in Noma management and the surgical campaign

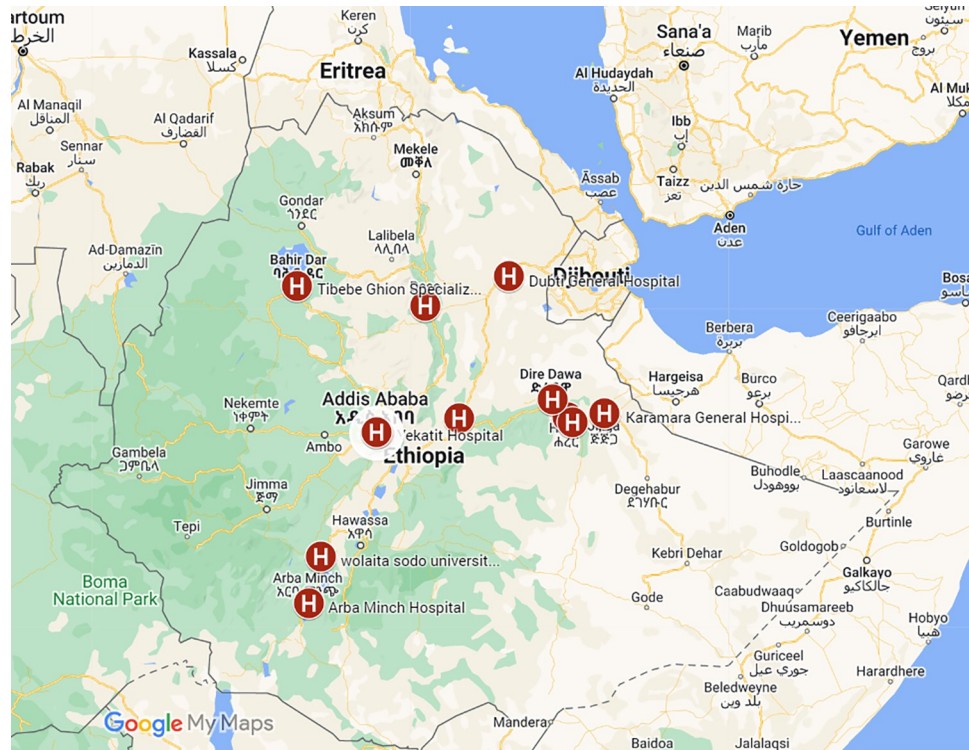

**Fig 1. Map of Ethiopia, distributions of the selected hospitals selected for rapid assessment of noma in September 2022 (Source: https://open.africa/dataset/ethiopia-shapefiles).**

and two country directors of noma-focused project. A combination of purposive and snowball sampling was used to select the interviewees, who were contacted in-person and via email, to ask for their consent to participate.

All interviews were conducted in-person in English and recorded in writing. An interview guide with open-ended questions was developed to capture the experts' experience in noma

**Table 1. Regional and referral hospitals in Ethiopia selected for rapid assessment of noma in September 2022.**

| | Region | Town | Hospital | Catchment population |
|---|---|---|---|---|
| 1 | Amhara | Bahir Dar | *Tibebe Ghion specialized teaching hospital | 3 to 5 million |
| | | Boru Meda | Boru Meda specialized hospital | 1.5 million |
| 2 | Afar | Dubuti | Dubti referral hospital | 1.5 million |
| | | Amibara | Mohammed Akile memorial general hospital | 1.5 million |
| 3 | Somali | jigjga | Karama general hospital | 1.5 million |
| | | | Jigjga University teaching hospital | 3 to 5 million |
| 4 | Southern nation nationality people (SNNP) | Arba Minch | Arba Minch general hospital | 1.5 million |
| | | Sodo | Wolayita otona referral hospital | 3 to 5 million |
| 5 | Harari | Harar | Hiwot Fana teaching hospital | |
| 6 | Oromia | Bisidemo | Bisidemo general hospital | 1.5 million |
| 7 | Addis Ababa | Addis Abeba | *Yekatit 12 referal Hospital and **Myungsung Christian Medical Center (MMC) | 3 to 5 million |
| 8 | Drie Dawa | Drie Dawa | Dilechora general hospital | 1.5 million |

*Hospitals with capacity for noma surgery

** Private Hospital

care, including the number of cases managed, role of partner organization, and the type of capacity building necessary to enable those institutions. Additionally interview explore existing strategies policies and plans for noma prevention and care; health sector capacities in service provision, information system, financing, human resource and resource needed in case management. Probing questions were included to further investigate the challenges and recommendations posed by experts and NGO's working in noma prevention and care in Ethiopia.

The interview note ware edited for clarity and transcribed. The qualitative analysis was conducted using Atlas.ti Version 23.4.0 (29342), employing a combined inductive and deductive thematic analysis approach with open coding.

Desk review: To consolidate insights from the qualitative and quantitative studies, a desk review was conducted, synthesizing existing literature,-both peer-reviewed and unpublished on noma case report or disease burden studies in Ethiopia. The review also examined health policies, curricula for health professional educations, oral care guidelines, and the current status of case reporting system.

The desk review involved searches in academic databases such as Pubmed/Medline, Web of Science and google scholar, as well as resources from the Ethiopian Ministry of Health and NGO's databases. Search terms included "Ethiopia," "noma," "health," "oral health," "cancrum oris", "gangrenous stomatitis", "gingivitis," and "neglected disease," ". Additionally, oral health experts, the Directorate of Disease Prevention and Health Promotion, neglected Tropical Disease focal person and NGO's engaged in noma care were contacted for additional resources. Literatures and documents in both English and Amharic were reviewed, with findings organized in thematic areas, including disease burden, policy frameworks for noma prevention and care and health system reediness.

## Result

### Stakeholder analysis and desk review

**Stakeholder analysis.** *The Ethiopian MOH and the Regional health bureaus are the main players of the health system in designing a policy, developing strategic plans, budgeting and cascading implementation of any health program including Noma.*

In the stakeholder analysis the assessment found two hospitals, two NGOs and one rehabilitation center actively involving in Noma case finding, clinical care and rehabilitation.

*Notable Stakeholders*:

- Project Harar: is a charity registered in England and Wales and in Ethiopia. At the binging (2003) the organization was helping noma patients in getting surgery in UK but letter on the project expanded to any facial disfiguring diseases which needs surgical intervention. Since 2004 the organization has developed a unique outreach model to find patients from even the most remote rural areas, and to transport them to Addis Ababa for the surgery. Project Harare is working specifically with Yekatit 12 Hospital in Addis Ababa organize the surgical campaign.

- Facing Africa: A UK Registered charity with office in Ethiopia started working in 2007 in noma care and other facial deformities which need surgical intervention. Since 2007 Facing Africa fund and organises two bi annual surgical trips to Ethiopia independently. The main activity of Facing Africa is in fund raising to support teams of professionals each year from Europe to Ethiopia to cover air fares. The organization also uses local experts (plastic surgeons) and collaborate with two Hospitals in Addis Ababa (Yekatit 12 Hospital and Myungsung Christian Medical center) to perform the operation

*Key Hospitals*:

- Yekatit-12 Hospital, Addis Ababa: is a public teaching hospital in Addis Ababa and is a key partner in providing noma and cleft surgery for Project Harar's patients and Facing Africa. Each year the hospital hosts one to two complex surgical missions. UK, international and Ethiopian medical staff work together at Yekatit-12 to provide surgery to patients with complex facial conditions.

- Myungsung Christian Medical center (Korea Hospital): is a missionary general hospital in Addis Ababa. The hospital offers plastic surgery and maxillofacial surgery. The hospital works with Facing Africa in providing surgical service for noma patients

- Tibebe Ghion Specialized Hospital: a public teaching Hospital of Bahir Dar University, in Amhara region, Bahir Dar. The Hospital has a head and neck surgeon who is performing noma surgery integrated in the routine care (Fig 2).

*Policy*: The Federal Democratic Republic of Ethiopia put in place the National Health Policy and Strategy in 1993. The development of Health Sector Development Plan (HSDP) of the National Growth and Transformation Plan (GTP) and Health Sector Transformation Program (HSTP) implemented the policy throughout the country. In the last two decades different initiatives including hospital reforms has been implemented to improve services as well as pro-poor interventions to reorient health services towards health promotion, disease prevention, curative services throughout the country [19].

Because of these initiatives currently the country is well placed in achieving primary health care coverage. Ethiopia has achieved substantial progress in improving many key health indicators during the past two decades.

●

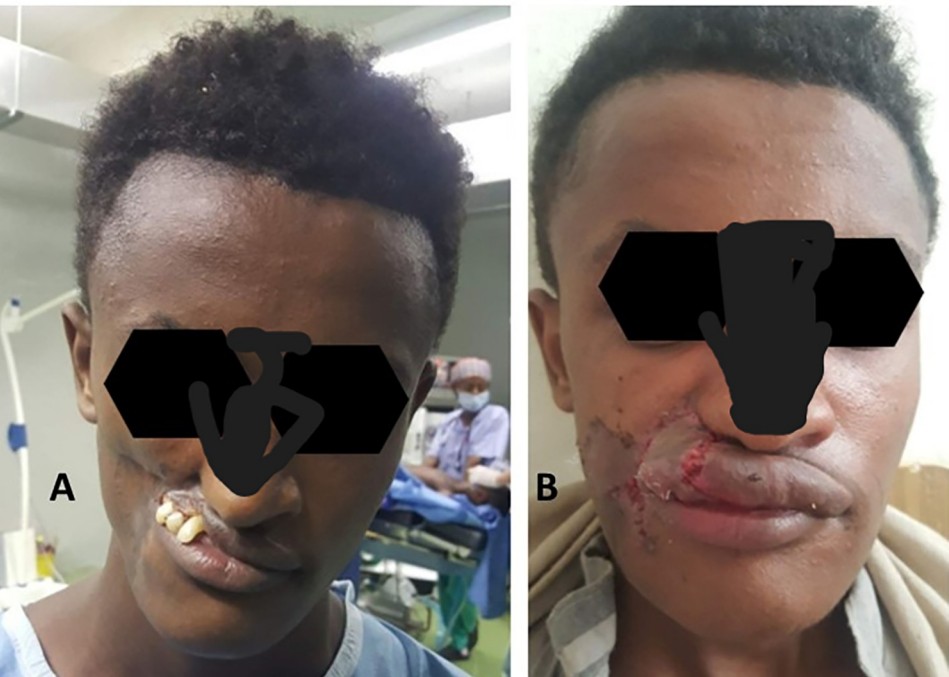

**Fig 2. A 22 year old male patient (before surgory and two weeks after surgory), surgory performed in the routine program in Tibebe Ghion Specialized Hospital, Bahir Dar, Ethiopia (Photo courtsy of Melesse Gebeyehu, 2022).**

The major health problems of the country remain largely preventable communicable diseases and nutritional disorders. Malnutrition and WaSH, are considered as the top two risks that drove the most deaths and disabilities in the country [18].

There is no standalone national policy on noma in Ethiopia. However, the health policy of the country emphasizes the prevention and control of diseases of poverty.

*Health work force*: In the country health professional training initiative dental care program was expanded from one public centre in Addis Ababa to multiple both public and private schools both in Addis and in the regions. The programs delivers training for Dental professionals, Dental surgeons and maxillofacial surgery. But there is no program for training of dental hygienist/therapist or dental practitioner/nurse. The last national human resource strategic plan (2016–2025) in its health facility standard projected to assign at list one dental hygienist at health canter level to prevent and treat oral diseases as well as to promote oral health need. Despite all the last report (2016) the number of dental professional including dentists and dental hygienists working in the country health system was only 270. The strategic plan has also projected to assign minimally two dental professionals (BSc) to primary hospital, but the assessment could not find record or other evidences on the implementation of the plan [19]. The Inservice curriculum for Dental doctors includes noma and its prevention and management. Inservice training Dental care training is also included in doctor of medicine and Nursing curriculum with one credit hour. Currently there is no pre-service training or continuous professional development for oral/dental care in the country.

*Health information system*: The Ministry of Health has also put in place HMIS to gather data from all health facilities in the country. The system is in its scale-up phase and need to be expanded to address diseases of the poor. Currently the system does not include the indicators to acute cases and sequel of noma.

*Literature reviews*: The first report of noma cases in Ethiopia who has received reconstructive surgery from medical mission (project Harar) was from 2007 to 2009 which has reported 43 cases. The report emphasized the similarity of the gross disease encountered to that faced by surgeons in the nineteenth century. The article has also advocates for teams of healthcare workers to move to areas of need rather than sending patients abroad for the surgery [15]. The second report by another medical mission (Facing Africa) was a retrospective review of 81 cases who has undergone reconstructive surgery from 2015 to 2019. The report uncovered critical information's such as age of onset (5 year and 8 months), median time of presentation (18 years from onset to accessing treatment), geographic distribution of patients, and significantly decreased patient quality of life [14].

A retrospective study to assess the clinical presentation of patients in 161 noma patients registered in three centres in Addis Ababa. The facial defects ranged from minor to severe tissue damage with 19.3% grade 1 (0–25% tissue damage), 42.3% reported as grade 2 (with 25 to 50% tissue damage), 30.7% grade 3 (50 to 75% tissue damage) and 8.1% grade 4 (75 to 100% tissue damage). The commonest anatomic region affected by the disease are cheek, upper lip, lower lip, nose, hard palate, maxilla, oral commissure, zygoma, infra-orbital region, mandible, and chin [20,21].

## Health care workers knowledge and health sector capacity assessment

Forty federal and regional health system leaders, experts, and NGOs who are working on NTDs responded for the rapid assessment on knowledge, practice and Information about noma. Among the participants, 27 were public health professionals and specialists, nine medical doctors, two laboratory technologists, two health care administrators. The mean experience in the health system was 18 years (ranging 2 to 45 years). The participants' roles varies from NTD experts and researchers to directors working in public sector and NGOs.

Fifteen (37.5%) participants reported having heard about noma, but only 6 (40%) of those with prior information have seen a case. Eighty-five percent of the participant stated that there is no oral health program in the country's health system. All participant agreed on the limitation of on the availability of human resource for oral health and poor capacity of the health institution for integrating of oral health program in to the primary health care unit.

Sixty percent of respondents said noma can be integrated into NTD programs, while 40% preferred integration into Non Communicable Disease, nutrition and medical services.

For the key informant interviews, three plastic surgeon with 10 to 15 years of experience working on noma were selected. Two NGO's country directors, who have been working on noma for 5 to 7 years, were also interviewed. A plastic surgeon performing constructive surgery stated, "In the last 10 years, I have seen about 300 cases, but since 2020, we are seeing fewer noma cases. However considering the country situation with war and drought, it is expected that noma cases will increase in the coming years. We have not operated on noma patients for the past two years due to COVID 19 pandemic movement restriction.".

Regarding noma prevention and the surgical intervention, another plastic surgeon responded, "Noma prevention activities do not exist until now, but it is very important to initiate and develop a prevention system, especially at this time, because the ongoing war and drought will likely increase the number of noma patients. Prevention should focus on both primary and secondary levels to prevent the occurrence of the disease and minimize complications and suffering. Activities should include supporting existing noma management centers, establishing new centers at different levels, creating awareness, training professionals, procuring instruments, and securing financial resources to support and upgrade available centers.

"The surgery for noma is demanding because the destruction caused by the disease is usually extensive and involves vital body parts. The patients' nutritional status is typically very poor, and the surgery is comprehensive, staged, demanding, and requires the expertise of various professionals. To achieve good surgical outcomes, patients need preoperative and postoperative rehabilitation, including nutritional, psychological, physical, and occupational therapy. Unfortunately, these services are not available in the routine health system."

A head and neck surgeon from Bahir Dar with experience in performing complex facial tumor and noma surgeries stated, "We have performed some stage 5 noma surgeries in the routine system by Ethiopian surgeons with successful outcomes. We are training surgeons at St. Paul Millennium Hospital and Medical College to perform these surgeries" (Fig 2).

A key informant from an NGO working on recruiting patients and supporting public hospitals for surgical management, when asked about the type of capacity building necessary, stated, "Training for plastic surgeons (residency) and equipment are the main areas of support. Due to the complicated nature of the surgery, postoperative services are very important." Regarding the main challenges faced by NGOs working on noma patient care, they noted, "Equipment, pre- and post-operative care and treatment accommodation, medication, human resources, timely approval of medical team credentials by the Ethiopian Food and Drug Authority, ongoing training for the local team on case management, easy and affordable/subsidized diagnostic centers, follow-up of cases after discharge, and the availability of pre- and post-operative accommodation centers for patients and parents. Identifying cases is also a challenge that I encountered during my time on the project."

## A retrospective assessment of noma cases

The retrospective identified 69 reported case of noma from January 2015 to December 2020, with 97% of cases documented by two NGOs supporting surgical mission (Fig 3). 38 (55.1%) of cases are female and 44.9% of cases are male. The mean age of patient is 28 years (ranging 4

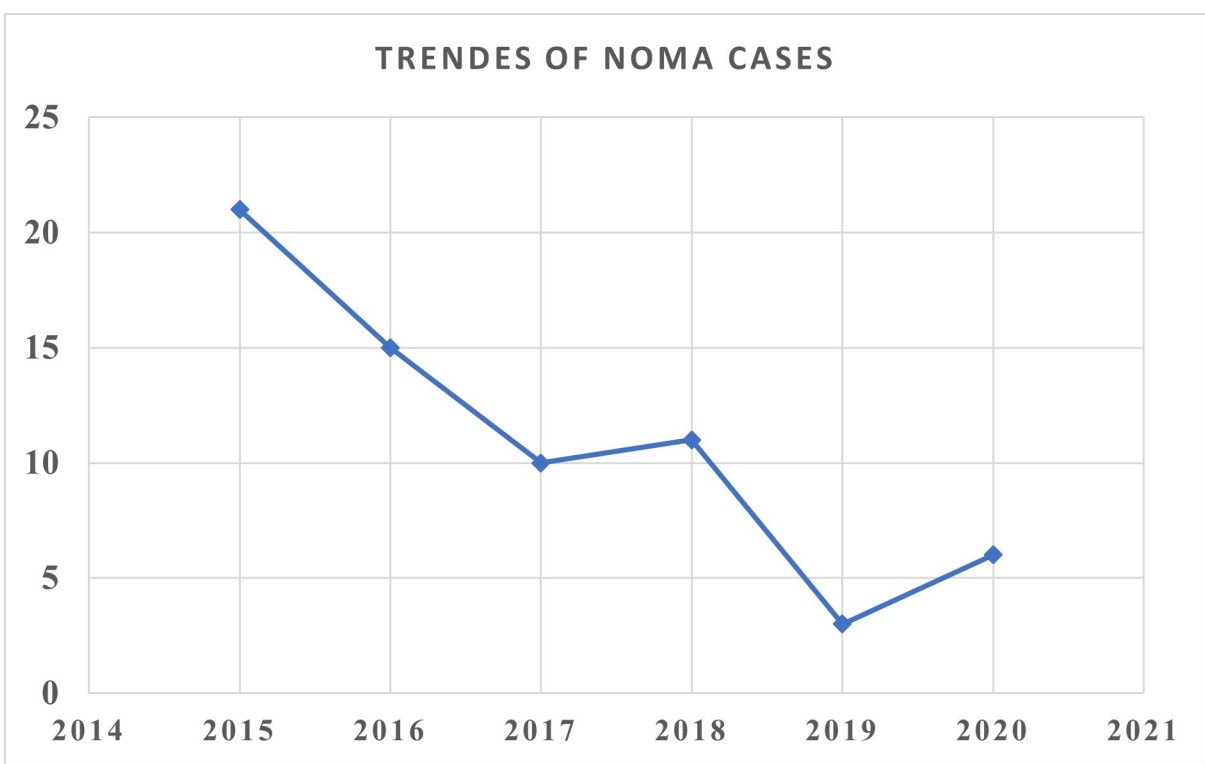

**Fig 3. Trend of noma cases who have undergone reconstructive surgery in Ethiopia from 2015 to 2020.**

to 70 years). The recorded patient address indicated that 31 patients are from Amhara region, 9 from Somalie, 8 from Oromia, 5 from Afar, 5 from SNNPR, 4 from Gambella and 3 from Addis Ababa (Fig 4). Most of the report (67 patients) were extracted from the NGO data base who are working on noma care. Two patients the surgery was performed by in Bahir Dar by Ethiopian Head and neck surgeon in the routine care program without the assistance of the NGOs.

Two of the patients with address in Gambela region are from Sudanese refugee camp. There is no record for patient clinical presentation, duration of illness, severity and patient outcome.

A study from Ethiopia on the risk factor has also concluded poverty-related diseases such as malaria, helminths infection, measle, diarrheal diseases, and unfavourable living conditions were identified to be the risk factor for noma. As such the disease is truly preventable. Prevention of the disease can be achieved through promoting overall awareness of the disease, poverty reduction, improved nutrition, and promotion of exclusive breastfeeding in the first 3–6 months of life. Furthermore, optimum prenatal care, timely immunizations against common childhood diseases, initiating vaccination, and improving the social living conditions are the other preventive mechanisms. Moreover, long-lasting economic development should be considered to effectively and sustainably prevent the disease.

The mean self-reported age of onset was 5 years and 8 months, with a median time of 18 years from onset to accessing treatment. Before intervention, 65% covered their face in public, 59% reported difficulty eating, 81% were unhappy with their appearance, and 71% experienced bullying.

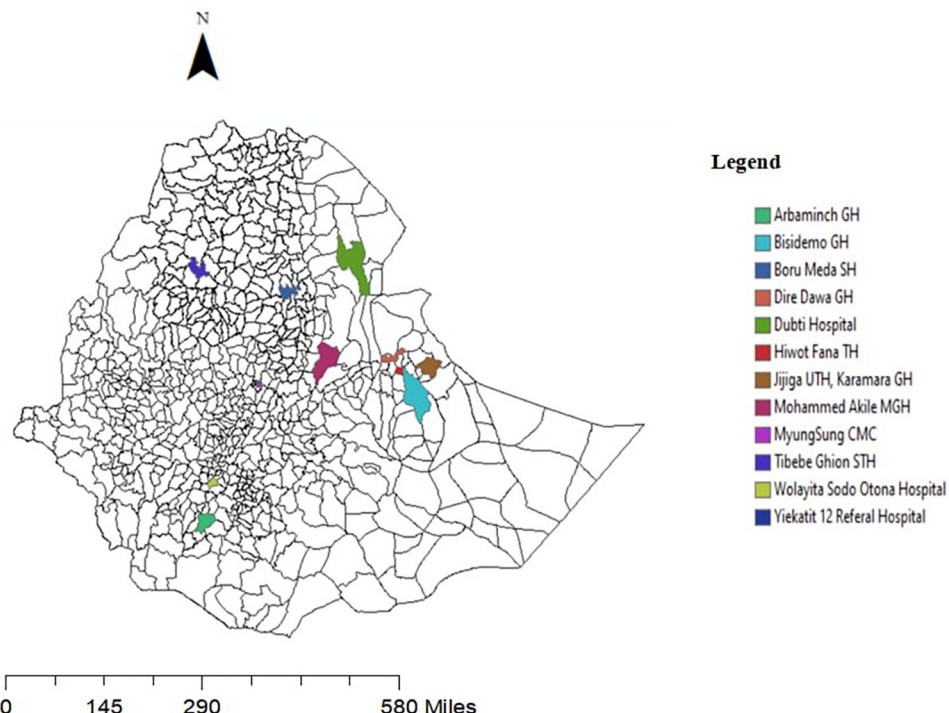

**Fig 4. Geographical distribution of noma cases who have undergone reconstructive surgery in Ethiopia from 2015 to 2020 (https://open.africa/dataset/ethiopia-shapefiles).**

## Discussion

The assessment indicated that noma is endemic in Ethiopia, with cases reported from all regions of the country and regional variation in case distribution. The highest case coming from Amhara region (45%) followed by Somali region (13%). The strong focality of noma cases in Amhara and Somali region also goes also with the highest prevalence of childhood malnutrition on those regions [21]. The lack of records for acute cases, calls for urgent development of epidemiological surveillance programs and improving the HMIS/DHIS indicators to accurately document noma. More accurate data are needed and for this it is important to develop a routine program for identifying and recording cases of noma at the primary health facility level.

The assessment revealed a declining trend in the number of noma cases receiving care from 2015 to 2020. However, as most of the surgeries are conducted through missionary-supported campaigns, this trend may not reflect the true incidence of noma. Previous studies in the country showed that the mean age of onset of 5.6 years and a median age at presentation 25 years, which is in agreement with our findings. The median delay between disease onset and access to treatment of noma patient in the country is 18 years [14], reflecting limited access to specialist health care in the region. Additionally, the assessment have identified only three hospitals-two in Addis Ababa and one in Bahir Dar- that are managing noma patients and the hospitals in Addis Ababa provide service only during a medical mission. Studies on the clinical presentation of noma patients in the country showed that most of the patients suffered from extensive noma-induced facial disfigurements, and functional deficit with significant social discrimination, and negative psycho-social effects [14,20].

A study on the risk factor of noma in Ethiopia recommended that prevention of the disease can be achieved through promoting overall awareness of the disease, poverty reduction,

improved nutrition, promotion of exclusive breastfeeding in the first 3–6 months of life, providing optimum prenatal care, timely immunizations against common childhood diseases, initiating vaccination, and improving overall living conditions. Long-term economic development is also essential to sustainably prevent the disease [20].

In the last report of Human resource(HR) for health (2016) in Ethiopia indicated only 270 dental professionals in the national healthcare system, equating to roughly one dental professionals per 300,000 people. While the HR strategic plan projected an increase to 6576 dental professionals in 2020 and 10,017 by 2025, implementation of this plan is not documented [19]. Besides the limited distribution of oral health personnel, Ethiopia's primary healthcare system lacks adequate equipment and functional facilities, leaving most of the population with minimal or no access to professional oral health care services. This results in a high proportion of untreated oral diseases and significant needs and demands for essential oral health care services, thus posing challenges to primary health care systems in the country.

The strength of the study is that, this is the first study to document regional distribution of noma cases in Ethiopia. The study also analysed the health system strength and gap which will inform policy makers for noma prevention and control program in the country. However, the assessment faced some limitations, including difficulty accessing information from hospitals and NGOs working on noma and lack of source documents at a missionary hospital performing noma surgeries. Existing records contained limited information on individual noma patients, and there were no corroborating records of cases reported by physicians or acute cases documented in the hospital HMIS. The assessment did not thoroughly evaluate existing laws, regulations, policies, and strategies addressing noma.

## Conclusion and Recommendations

The assessment demonstrated that noma cases are reported from all regions in Ethiopia, with regional variation in case distribution. The absence of record for acute cases underscores the urgent development of epidemiological surveillance programs and improving the HMIS indicators to document noma case accurately. More accurate data require a routine program for identifying and recording cases of noma at the primary health care level.

This rapid assessment has identified a striking scarcity of research and surveillance programmes for noma. Addressing noma effectively will require its inclusion in the national list of neglected tropical diseases and the implementation of integrated control programs aimed at elimination.

There is a need to intensify public awareness effort and enhance education for community workers and healthcare professionals at primary health care level on identification of noma with emphasis on recognizing each clinical stages and insuring timely referral to specialized hospitals.

Considering the country recent challenges, including war, the pandemic, drought and health system disruption-as well as the interruption of medical mission, active surveillance is needed to identify both acute cases and individuals with scaring disfigurement requiring reconstructive surgery.

We recommend future research that includes data collection at the district health facility level to determine the true burden of noma. Prospective studies are also necessary to understand the sequence of events that contribute to the development of noma.

Given the limited number of dental professionals in Ethiopia, it is impractical to rely solely on dental workforce models to deliver professional treatment at primary healthcare level. An opportunity lies in leveraging the existing health extension program, which is a key component of the primary health system workforce. Expand oral health services by integrating them into

school health programs and the existing NCD or NTD prevention and control programs in the primary health system are also another opportunities to explore.

By following these recommendations, it will be possible to develop effective policies and programs that address noma, improve oral health services, and ultimately contribute to the elimination of this neglected disease.

## Supporting information

**S1 Data. Database for National noma cases who have undergone reconstructive surgery in Ethiopia from 2015 to 2020.**
(XLSX)

## Acknowledgments

Ministry of Health of Ethiopia for the guidance and administrative support

Project Harar and Facing Africa for sharing data of the Noma cases supported by their mission

Dr Mekonnen Eshete and Dr Melesse Gebeyhu, for their expert input in to the inception of the assessment, providing me information and contact of the stakeholders and their guidance in understanding the Noma surgical care delivery in the country.

To Alemayehu Bekele for his hard work on field based data collection from the selected institution.

And special thanks to WHO country office (Dr Zeyede Kebede, Dr Nigus Manaye, Dr Worku Bekele) and WHO Regional Office for Africa (Dr Yuka Makino) for their guidance, expert advice from the inception to finalization of the field work.

And to all those who directly or indirectly gave their contribution to this work.

## Author Contributions

**Conceptualization:** Wendemagegn Enbiale.

**Data curation:** Wendemagegn Enbiale.

**Formal analysis:** Wendemagegn Enbiale.

**Investigation:** Wendemagegn Enbiale.

**Methodology:** Wendemagegn Enbiale.

**Project administration:** Wendemagegn Enbiale.

**Validation:** Wendemagegn Enbiale.

**Visualization:** Wendemagegn Enbiale.

**Writing – original draft:** Wendemagegn Enbiale.

**Writing – review & editing:** Wendemagegn Enbiale.

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
