## [Decision Letter · Decision Letter 0]

29 Mar 2024

Dear Dr Enbiale,

Thank you very much for submitting your manuscript "Rapid assessment of Noma: Reporting on forgotten and neglected disease in Ethiopia" for consideration at PLOS Neglected Tropical Diseases. As with all papers reviewed by the journal, your manuscript was reviewed by members of the editorial board and by several independent reviewers. In light of the reviews (below this email), we would like to invite the resubmission of a significantly-revised version that takes into account the reviewers' comments. 

We cannot make any decision about publication until we have seen the revised manuscript and your response to the reviewers' comments. Your revised manuscript is also likely to be sent to reviewers for further evaluation.

Sincerely,

Elsio A Wunder Jr, DVM, Ph.D.

Section Editor

Reviewer's Responses to Questions

**Key Review Criteria Required for Acceptance?**

**Methods**

-Are the objectives of the study clearly articulated with a clear testable hypothesis stated?

-Is the study design appropriate to address the stated objectives?

-Is the population clearly described and appropriate for the hypothesis being tested?

-Is the sample size sufficient to ensure adequate power to address the hypothesis being tested?

-Were correct statistical analysis used to support conclusions?

-Are there concerns about ethical or regulatory requirements being met?

Reviewer #1: The methods chapter of this manuscript is one of the weakest sections. There is an overarching objective of the study, but no specific objectives or hypotheses. 

The author(s?) used many different methodologies, which are contradictory in the abstract, methods and results section of this paper. I am missing a clear description of each method used. For the desk and literature review (what is meant by desk review?), there is no information on the type of literature reviewed, where the literature came from, how it was reviewed, how data was extracted and analysed, and if there were any inclusion or exclusion criteria. 

For the health sector capacity assessment, there is no information what this means and how they performed it. Furthermore, for the cross-sectional retrospective assessment, information about statistical procedures is only given for hospitals but not NGOs and information about the type of data that was collected is not complete. There is also a qualitative methods section (missing in the study design) which does not give any information on sampling strategy of informants, how the interviews were conducted and by whom. In addition, there is no information about how the interviews were processed (transcription? analysis?). 

The rapid knowledge assessment, found in the results is not at all described in the methods.

Reviewer #2: The overall purpose and objectives of the paper have not been clearly articulated in the abstract. In terms of the objectives, perhaps some more clarty could be offered about the what exabley is meant by stakeholder engagement and the word national should be put before health system. 

It it my opinion that the information under general setting should be embedded into the introduction somewhere

With resepct to the data collection and analysis, 1. While it is commendable that the health institutions are listed, the rationale for chosing these centers should have been placed before the listing. Secondly, as an international reviewer, the listing means nothing to me. Rather, it would have helped if a map was provided that showed the locations of these health institutions. To my mind, if these institiutions are spread accross the country, then this would give a clear indication of geographical representation. Furthermore i am left asking if the instititon had to meet all of the inclusion criteria or not. If they did not have to meet all of the inclusion criteria, then perhaps a table could be generated to show which criteria were met for each institution. 

In terms of the qualitative assessment, some details should be provided about the instrument used in the semi structured interviews. Furthermore, no details about the interviews conducted with the plastic surgeons were provided

Reviewer #3: The objectives of the study are clearly articulated. Notwithstanding the aforementioned and the inherent scientific-social value of the study, weaknesses are observed regarding the methodological aspects of the research. It concerns a study with qualitative-quantitative methodologies, and I consider it important that this aspect be made clear, even in the structuring of the writing. When referring to the 13 Hospitals, it is relevant to specify the management type of the healthcare facility (whether public, private, philanthropic, or mixed). Regarding the examined medical records and considering it is a retrospective study, the sample is appropriate and simple descriptive statistical data were presented. However, concerning the collection of qualitative data, validity cannot be ascertained since the eligibility criteria of the professionals who answered the questionnaire were not presented, and the ethical approval from the Federal Ministry of Health of Ethiopia was deemed unnecessary. I advice the removement of reference 20 and all related excerpts, in sight of the reference not been reviewed or published yet.

**Results**

-Does the analysis presented match the analysis plan?

-Are the results clearly and completely presented?

-Are the figures (Tables, Images) of sufficient quality for clarity?

Reviewer #1: The results chapter is extensive and interesting. However, it is missing a clear structure and is too long. For some results, especially health systems- and policy-related results, it is difficult to comprehend how they were collected (probably due to lack of clarity in the methods chapter). 

Right now, the biggest focus lies on the health system and policies. From an epidemiological point of view, I would recommend to shift the focus to the number of noma patients identified, their socio-demographic and disease-related information, the survey and perceptions of health care providers. The health systems paragraph could be much shorter, including tables that give information about policies and health facilities treating noma.

The qualitative results section is mainly made up from three different quotes. I recommend to first give information about your respondents (all of them) and then to paraphrase your results in paragraphs according to topics (e.g. one about the noma cases practitioners have encountered, one about needs they see, one about potential solutions). You can still use quotes, but try to limit them to 1-3 sentences.

Figures are not always clearly mentioned in the text and captions are missing in the appendix. Additionally, the figure quality could be improved.

Reviewer #2: I note that there are findings presented from the stakeholder analysis, but the details of how the stakeholder anaylysis was conducted appears not to be present in the method. 

I see information about a KAP in the the results, but there is no description in the method about a KAP and none of the study's objectives speak to a KAP. 

Refer to previous comment in method about lack of details on the conducting of the interview, but I see results from the interview in the results section.

Reviewer #3: For the study proposal, the presented results are comprehensive. However, they mainly derive from studies already described in the scientific literature. Nonetheless, this fact does not diminish the importance of the study, nor does it render its publication unfeasible.

**Conclusions**

-Are the conclusions supported by the data presented?

-Are the limitations of analysis clearly described?

-Do the authors discuss how these data can be helpful to advance our understanding of the topic under study?

-Is public health relevance addressed?

Reviewer #1: The conclusions are supported by the presented data. Limitations are described in the discussion chapter, however, it is difficult to judge if the main limitations of the study are mentioned because the methods chapter is lacking clarity. 

The author(s?) give many great recommendations for an improvement of the health system in order to improve noma surveillance, prevention and treatment. The discussion could still gain some more clarity about how the results advance the understanding of noma in Ethiopia and what public health relevance (also internationally) these results have.

Reviewer #2: The limitations that are presented in my opinion, should have been key information that should have been presented early on in the results. I believe the use do the word endemic is incorrec in this context. The second sentence in the conclusions and recommendations needs urgent revision

Reviewer #3: The conclusions presented were well-discussed based on the data provided. The study limitations were outlined, and the advancements in scientific and social knowledge that the study may bring about were discussed. The importance of the work for defining strategies and public policies capable of benefiting the affected population and reducing suffering of all kinds is emphasized.

**Editorial and Data Presentation Modifications?**

Reviewer #1: In general, this manuscript would profit from a clear structure within the different chapters. In addition, there are some additional editorial recommendations:

- Authorship: I have almost never encountered any public health publication with only a single author. Did you not receive any support in the study design, data collection, analysis or writing of this paper? If not, please clarify this by writing “I” instead of “we” in the manuscript chapters. If yes, please invite co-authors.

- active/passive voice: it is nice for the reader if you make a clear decision about using only active or only passive voice

- Please review the capitalisation of words. Noma is usually written in small letters …

- Abbreviations: You use many abbreviations, do not forget to introduce them for the first time.

- Figures: In the text, figure 5 appears before figure 1. Make sure, your numbering is coherent.

- Abstract: I recommend to shorten the background and better elaborate the methods and and results section. Additionally, the results section needs a clearer structure.

- Introduction: The introduction is very well-written and comprehensible. There is one minor point concerning the causes of noma: At the end of the second paragraph you write about the damage to the mucosa barrier and that the bacterial factor remains to be unidentified. This is partly true. According to the latest knowledge researchers think that due to noma risk factors (such as malnutrition and weakening of the immune system), the balance of bacteria in the mouth changes leading to the onset of the disease.

- Latest prevalence estimates and historical reports: I recommend to read Galli et al. 2022: Prevalence, incidence, and reported global distribution of noma: a systematic literature review in the Lancet Infectious Diseases. If you need access to the PDF, let me know.

- Setting/ data collection: In the data collection chapter, you nicely name all of the facilities you worked with. I think you could even add this to the setting and just produce a map instead of listing everything in the text.

- Page 12: Most of the last paragraph belongs to either the introduction or the discussion. 

- Limitations: usually, studies also have strengths � I recommend a chapter called strengths & limitations.

Reviewer #2: General formatting needs to be examined. The Referencing style also needs to be re-examined as inconsistencies were noted.

Reviewer #3: (No Response)

**Summary and General Comments**

Reviewer #1: It is great to see that the interest in noma research is starting to increase. Studies examining noma occurrence and related national policies are crucial to advance knowledge on the epidemiology of noma. So far, literature about noma prevalence is very scarce and it is difficult to have access to patient records of regions where noma is suspected to exist. Therefore, I think that this manuscript is highly relevant in making data on noma in Ethiopia accessible to the international community. In addition, the authors analyse how the health system needs to be adapted in order to start noma prevention and surveillance activities and to increase noma treatment capacity. This could be very useful information for other countries in similar contexts. 

Despite the great thematic angle and importance of this paper, clear objectives are missing and the methods chapter needs to be re-written. In the current state, it is impossible to judge the data quality due to a lack of transparency and detail in the methods chapter. In addition, a clearer structure throughout the paper needs to be established.

Reviewer #2: In my opinion the study is relevant and the findings have potential to contribute significantly to the conversation on how resources are allocated to negleted tropical diseases, especially the rare ones such as noma. While the importance and timeliness of this paper is not being questioned, it is my recommendation that major revisions be done before this paper is accepted for final publication. As the wealth gap continues to expand gloablly with more people being pushed to margins of society, lacking access to basic needs, there is an elevated risk to observing a higher global incidence of noma.

Reviewer #3: (No Response)

PLOS authors have the option to publish the peer review history of their article (what does this mean?). If published, this will include your full peer review and any attached files.

Reviewer #1: No

Reviewer #2: No

Reviewer #3: No
---

## [Decision Letter · Decision Letter 1]

29 Oct 2024

PNTD-D-23-01530R1Rapid assessment of Noma: Reporting on forgotten and neglected disease in EthiopiaPLOS Neglected Tropical Diseases Dear Dr. Enbiale, Thank you for submitting your manuscript to PLOS Neglected Tropical Diseases. After careful consideration, we feel that it has merit but does not fully meet PLOS Neglected Tropical Diseases's publication criteria as it currently stands. Therefore, we invite you to submit a revised version of the manuscript that addresses the points raised during the review process.The reviewers were excited with the willingness of the authors to address all the reviewer's suggestions. They are just recommending to have the manuscript going through a thorough English grammar revision. Please submit your revised manuscript within 30 days Nov 28 2024 11:59PM. If you will need more time than this to complete your revisions, please reply to this message or contact the journal office at plosntds@plos.org. Please include the following items when submitting your revised manuscript:*
A rebuttal letter that responds to each point raised by the editor and reviewer(s). You should upload this letter as a separate file labeled 'Response to Reviewers'. This file does not need to include responses to any formatting updates and technical items listed in the 'Journal Requirements' section below.*
A marked-up copy of your manuscript that highlights changes made to the original version. You should upload this as a separate file labeled 'Revised Manuscript with Track Changes'.*
An unmarked version of your revised paper without tracked changes. You should upload this as a separate file labeled 'Manuscript'. If you would like to make changes to your financial disclosure, competing interests statement, or data availability statement, please make these updates within the submission form at the time of resubmission. Guidelines for resubmitting your figure files are available below the reviewer comments at the end of this letter. We look forward to receiving your revised manuscript. Kind regards, Elsio A Wunder Jr, DVM, Ph.D.Section EditorPLOS Neglected Tropical Diseases Elsio Wunder JrSection EditorPLOS Neglected Tropical Diseases

Shaden Kamhawi

co-Editor-in-Chief

Paul Brindley

co-Editor-in-Chief

 **Journal Requirements:** **Additional Editor Comments (if provided):****Reviewers' comments:** Reviewer's Responses to Questions

**Key Review Criteria Required for Acceptance?**

**Methods**

-Are the objectives of the study clearly articulated with a clear testable hypothesis stated?

-Is the study design appropriate to address the stated objectives?

-Is the population clearly described and appropriate for the hypothesis being tested?

-Is the sample size sufficient to ensure adequate power to address the hypothesis being tested?

-Were correct statistical analysis used to support conclusions?

-Are there concerns about ethical or regulatory requirements being met?

Reviewer #2: Yes to all questions

**Results**

-Does the analysis presented match the analysis plan?

-Are the results clearly and completely presented?

-Are the figures (Tables, Images) of sufficient quality for clarity?

Reviewer #2: Yes to all questions

**Conclusions**

-Are the conclusions supported by the data presented?

-Are the limitations of analysis clearly described?

-Do the authors discuss how these data can be helpful to advance our understanding of the topic under study?

-Is public health relevance addressed?

Reviewer #2: Yes to all questions

**Editorial and Data Presentation Modifications?**

Reviewer #2: No comment

**Summary and General Comments**

Reviewer #2: This is a much improved version from the previous submission. Kudos to the authors who made all of the recommended changes.

PLOS authors have the option to publish the peer review history of their article (what does this mean?). If published, this will include your full peer review and any attached files.

Reviewer #2: No

---

## [Editor Report · Decision Letter 2]

13 Nov 2024

Dear Dr Enbiale,

We are pleased to inform you that your manuscript 'Rapid assessment of Noma: Reporting on forgotten and neglected disease in Ethiopia' has been provisionally accepted for publication in PLOS Neglected Tropical Diseases.

Best regards,

Elsio A Wunder Jr, DVM, Ph.D.

Section Editor

Elsio Wunder Jr

Section Editor

Shaden Kamhawi

co-Editor-in-Chief

Paul Brindley

co-Editor-in-Chief

---

## [Editor Report · Acceptance letter]

7 Dec 2024

Dear Dr Enbiale,

We are delighted to inform you that your manuscript, "Rapid assessment of Noma: Reporting on forgotten and neglected disease in Ethiopia," has been formally accepted for publication in PLOS Neglected Tropical Diseases.

Best regards,

Shaden Kamhawi

co-Editor-in-Chief

Paul Brindley

co-Editor-in-Chief
